# Crossbred Sows Fed a Western Diet during Pre-Gestation, Gestation, Lactation, and Post-Lactation Periods Develop Signs of Lean Metabolic Syndrome That Are Partially Attenuated by Spirulina Supplementation

**DOI:** 10.3390/nu14173574

**Published:** 2022-08-30

**Authors:** Rosamaria Lugarà, Simone Renner, Eckhard Wolf, Annette Liesegang, Rupert Bruckmaier, Katrin Giller

**Affiliations:** 1Animal Nutrition, ETH Zurich, Eschikon 27, 8315 Lindau, Switzerland; 2German Center for Diabetes Research (DZD), Ingolstaedter Landstrasse 1, 85764 Neuherberg, Germany; 3Molecular Animal Breeding and Biotechnology, Department of Veterinary Sciences, Ludwig-Maximilian University Munich, Gene Center, Feodor-Lynen-Strasse 25, 81377 Munich, Germany; 4Animal Nutrition, Vetsuisse Faculty, University of Zurich, Winterthurerstrasse 270, 8057 Zurich, Switzerland; 5Veterinary Physiology, Vetsuisse Faculty, University of Bern, Bremgartenstrasse 109a, 3001 Bern, Switzerland

**Keywords:** dietary fat, dietary sugar, microalgae, gene expression, liver steatosis, skeletal muscle, insulin-like growth factor, pig model

## Abstract

Excessive dietary intake of fats and sugars (“Western diet”, WD) is one of the leading causes of obesity. The consumption of the microalga *Arthrospira platensis* (spirulina, Sp) is increasing due to its presumed health benefits. Both WD and Sp are also consumed by pregnant and breastfeeding women. This study investigated if gestating and lactating domestic pigs are an appropriate model for WD-induced metabolic disturbances similar to those observed in humans and if Sp supplementation may attenuate any of these adverse effects. Pigs were fed a WD high in fat, sugars, and cholesterol or a control diet. Half of the animals per diet group were supplemented with 20 g Sp per day. The WD did not increase body weight or adipose tissue accumulation but led to metabolic impairments such as higher cholesterol concentration in plasma, lower IGF1 plasma levels, and signs of hepatic damage compared to the control group. Spirulina supplementation could not reduce all the metabolic impairments observed in WD-fed animals. These findings indicate limited suitability of gestating and lactating domestic pigs as a model for WD but a certain potential of low-dose Sp supplementation to partially attenuate negative WD effects.

## 1. Introduction

Excessive energy intake in the form of a “Western diet” (WD), often combined with a sedentary lifestyle, is considered the leading cause of the obesity epidemic, which affects approximately 30% of the adult population in industrialized countries today [1]. Chronic excessive intake of dietary saturated fat, sugars, and cholesterol may lead to an alteration in the activity of transcription factors that results in a dysregulation of gene expression favoring intra-organ lipid accumulation and, in the long term, other metabolic disturbances [2,3,4]. Even before the onset of the metabolic syndrome, including visible changes in body weight (BW) and body composition, shifts in metabolic and inflammatory blood parameters point towards an unfavorable metabolic state, as observed in rodents and primates [5,6,7].

Excessive energy intake and obesity are also increasingly prevalent in pregnant and breastfeeding women [8]. Since pregnancy and lactation are generally characterized by complex changes in metabolism as well as by intense immune activity [9,10], unhealthy diets and obesity during these periods additionally strain the organism [11].

The filamentous blue-green microalga *Arthrospira platensis* (also known as spirulina, Sp) is rich in bioactive phytochemicals, such as essential fatty acids (FA), vitamins, polyphenols, carotenoids, tocopherols, and phycocyanin [12]. These are presumed to contribute to the many beneficial health effects reported for Sp, particularly its antioxidant, anti-inflammatory, hypolipemic, and hepato-protective effects [13,14]. In vivo studies showed that Sp exerts these protective effects by modulating the expression levels and activities of transcription factors, thus regulating downstream transcription of target genes [15,16]. Regarding the potential beneficial health effects, the utilization of Sp as a supplement in human nutrition, including pregnant and breastfeeding women, is becoming increasingly popular [17]. Still, to the best of our knowledge, the potential metabolic effects of Sp intake during pregnancy and lactation have not yet been investigated.

Because of ethical issues that would arise from studies in pregnant and lactating women, different animal models have been employed, with rodents representing the most frequently used model for human nutrition and metabolism [18,19,20]. However, studies in rodents present some limitations that prevent the translation of research results to humans [21]. The pig (*Sus scrofa*) can be considered an interesting alternative to rodent models and might help bridge the gap between classical animal models and actual human physiology. Pigs share many anatomical, physiological, genomic and metabolic similarities with humans [22,23,24]. Minipigs have been mostly used to develop novel porcine models for metabolic disorders, but they are far more sensitive to an obesogenic diet than humans, because of a genetic predisposition for obesity [18]. Domestic pigs do not have this genetic predisposition and might thus be biologically closer to humans than the minipigs. Studies in pregnant and lactating pigs may shed new light on translational research investigating the metabolic effects of maternal WD.

Therefore, in the present study, we investigated if a diet rich in saturated FA (SFA), sucrose, fructose, and cholesterol in comparison to a commercial pig diet may induce a detrimental, obesity-related phenotype in gestating and lactating domestic pigs and if low-dose Sp supplementation may attenuate these potential adverse effects. So far, no study has evaluated if low-dose Sp supplementation proportional per unit of BW to recommendations in human nutrition (max. 8 g/day) to gestating and lactating pigs fed a WD might effectively attenuate adverse WD effects. We hypothesized that (i) feeding a WD to domestic pigs induces an obesity-like metabolic phenotype similar to that observed in humans; (ii) the effects of the WD differ during pre-gestation, gestation, lactation, and post-lactation periods; and (iii) low-dose supplementation of Sp may attenuate or even prevent the unfavorable phenotype observed in pigs fed a WD.

## 2. Materials and Methods

### 2.1. Animals, Housing, and Experimental Design

The experiment was approved by the Cantonal Veterinary Office of Zurich, Switzerland (license number ZH157/18), and performed in accordance with the Swiss legislation on animal rights and welfare. Lugarà et al. [25] provide a detailed description of the experiment. In brief, 24 female Large White × Landrace pigs (initial BW: 119.3 ± 8.2 kg, mean ± standard deviation (SD); average age: 5.6 ± 0.8 months; same sire; obtained from as few litters as possible to reduce the genetic diversity also from the maternal side) were housed in the Metabolic Center of the AgroVet-Strickhof research station (Lindau, Switzerland). Despite years of intense selection for leaner meat, when compared to other breeds such as Pietrain, such crossbreeds still possess genetic traits that allow for an increased intramuscular fat accumulation [26] and may thus also be susceptible to a WD. Due to limited housing capacity, the experiment was conducted in two identical runs (12 sows each, spring to winter in 2019 and 2020, respectively). The experimental design and schedule are outlined in Figure 1. Animals were housed in pairs at 17–22 °C and 60% relative humidity and fed ad libitum via automatic feeding stations (Pig Performance Tester, Nedap, Groenlo, The Netherlands), which recorded individual daily feed intake. Water and compressed straw were accessible ad libitum. After 6–8 weeks of pre-gestational feeding, animals were cycle synchronized via 18 days of oral application of Regumate^®^ (MSD, New Animal Health, Wellington, New Zealand). Artificial insemination (AI) was performed with fresh sperm from the same boar (PREMO^®^ Large White, Suisag, Sempach, Switzerland) for all animals. Animals were controlled for pregnancy on days 18 (heat detection) and 25 (ultrasonography) post AI. Animals not pregnant on day 25 were excluded from the experiment as further described below. From day 44 of gestation until weaning (30.4 ± 0.8 days after farrowing), animals were kept in individual farrowing pens. For weaning, the sows were moved back in pairs to the initial pens where they stayed for another 5 weeks until being slaughtered.

### 2.2. Experimental Diets

All diets used in this experiment were produced by Weinlandmuehle Truellikon (Glanzmann AG, Truellikon, Switzerland). For an initial adaption period of 2 weeks, all animals were offered a commercial compound feed designed to cover the basic nutritional requirements for gestating pigs. This diet was also used as the control diet (CTR_G_) during the whole pre-gestation, gestation, and post-lactation periods (Table 1). After the adaption period, animals were assigned to two dietary groups (n = 12 per group). Initially, sisters sharing the same mother were randomly distributed into the two groups. The remaining animals were randomly assigned, but groups were balanced for a similar average BW. All animals were fed ad libitum either the control diet (CTR group) or a WD (WES group) for an initial pre-gestation period of about 9 weeks before AI (Table 1, Figure 1). The WES_G_ diet contained 150 g/kg freshly fed fully hydrogenated palm oil (ALIkon, Erbo Spraytech AG, Buetzberg, Switzerland), 200 g/kg saccharose (crystalline sugar, Schweizer Zucker AG, Aarberg, Switzerland), 150 g/kg fructose (Fructofin C, Danisco, Kotka, Finland), and 2 g/kg cholesterol (cholesterol from sheep wool ≥92.5%, Sigma-Aldrich, Buchs, Switzerland). As intended, the WES feed contained substantially more ether extract (+450%), total sugars (+570%), and cholesterol (+1532%) and ultimately provided a higher gross energy content (+3.8 MJ/kg DM) than the CTR feed. In contrast, the CTR feed provided a higher starch content than the WES feed (+82%). The fiber content was slightly higher in the WES than in the CTR feed (8.4 vs. 6.5 g/kg DM). In terms of FA, the WES feed contained a higher proportion of total SFA (+262%) than the CTR feed. Contrarily, the CTR feed contained higher proportions of unsaturated FA. In spirulina, the major FAs were C16:0, C18:3 n–6 (γ-linolenic acid, GLA), LA, C16:1 n–7, and C18:0 (in descending order).

Feed allowance was 2 and 3 kg/day from day 40 to day 90 and from day 91 to day 105 of gestation, respectively. These restrictions were implemented to prevent problems at parturition that occur more frequently when sows are too heavy. One week before the expected parturition (day 107 of gestation), the diets of both groups were switched to lactation-type diets (CTR_L_, WES_L_) (Table 1, Figure 1). The lactation diets were fed ad libitum until weaning.

Two days after the final Regumate^®^ administration and thus 3 days before AI, daily manual supplementation with 20 g of spirulina (Sp) in tablets (IGV Planttech GmbH, Nuthetal, Germany) was started for half of the animals in both dietary groups (−: without Sp; +: with Sp). This ultimately resulted in four experimental groups, namely CTR−, CTR+, WES−, and WES+. The Sp supplementation was continued until slaughter.

### 2.3. Data and Sample Collection during the Feeding Experiment

Throughout the experiment, individual daily feed intake was determined on 2 days per week, either from data registered by the feeding stations in the pens or by recording the amounts of supply and leftovers in the farrowing pens. The BW was measured using an animal scale (Mettler-Toledo, Greifensee, Switzerland) at the start and end of each experimental period. Backfat thickness at the level of the last rib was monitored using a portable ultrasound instrument (Carometec, Frontmatec Kolding, Kolding, Denmark) at the same time points. Feed samples were collected at the start of each run and then every 8 weeks throughout the experiment for compositional analyses. One week before expected farrowing, total feces were collected over a period of 24 h to analyze fecal fat excretion. All samples were stored at −20 °C until further analyses.

### 2.4. Organ Collection at Slaughter

After 16 h of overnight fasting, the sows were transported within 30–45 min to the abattoir at Tierspital (University of Zurich, Switzerland). They were slaughtered on six different dates in random order by exsanguination after electrical stunning. Liver and perirenal visceral adipose tissue (VAT) were collected warm and weighed. Samples of liver and abdominal muscle (*Obliquus externus abdominis*) were snap-frozen in liquid nitrogen and stored at −80 °C until further analyses. Additional liver and muscle samples were vacuum-sealed in plastic bags and stored at −20 °C for FA analysis. Liver, muscle, and VAT samples were fixed in 4% paraformaldehyde prepared in phosphate buffered saline (both: Sigma-Aldrich) at 4 °C for 48 h.

### 2.5. Isolation and Quality Assessment of Liver and Muscle RNA

Total hepatic RNA was isolated after homogenization (MagNA Lyser Green Beads and MagNA Lyser Instrument (Roche, Basel Switzerland), 7000× *g*, 30 s) using the AllPrep DNA/RNA Mini kit (Qiagen, Hilden, Germany) according to the manufacturer′s instructions. To isolate total RNA from muscle tissue, samples were homogenized using TRIzol^TM^ Reagent (Thermo-Fisher, CA, USA) and the MagNA Lyser Instrument (6500× *g*, 25 s). After homogenization, chloroform (Sigma-Aldrich) was added, and samples were centrifuged (12,000× *g*, 10 min, 4 °C). The upper phase was mixed with 100% ethanol, and the RNA purification was performed using the RNeasy Mini Kit (Qiagen) according to the manufacturer’s instructions. The quantity of isolated RNA was estimated using a NanoDrop One spectrophotometer (Thermo Fisher, Waltham, MA, USA).

### 2.6. Transcriptome Analyses

Library preparation and RNA sequencing for liver and muscle tissue was performed by the Functional Genomics Center Zurich (Switzerland). For timing reasons, RNA sequencing was performed only for animals from the first run group, while qPCR was later used to validate the results in all animals (see next paragraph). Analysis of RNA integrity using a Fragment Analyzer System (Agilent Technologies, Wilmington, NC, USA) resulted in RQN numbers ≥8.6 for all samples. An RNA input of 1 µg was used for library preparation using the TruSeq Stranded mRNA Library Prep kit (Illumina, San Diego, CA, USA) following the manufacturer′s instructions. Quality-controlled libraries were sequenced in a single multiplex on a NovaSeq instrument (Illumina) in 100 bp single-end read mode. Overall, the sequencing depth was about 25 million reads per sample. Read alignment was executed using STAR [27] and the porcine genome (Sscrofa11.1) as reference. Raw read counts were calculated with the featureCounts option of the Rsubread package [28]. The analyses of differentially abundant transcripts (DATs) were performed in R (version 4.1.2, R Core Team, 2020) using the DESeq2 package (version 1.30.0, Michael Love, Boston, MA, USA). The analysis generated the *p*-values corresponding to a differential expression test and the log2 Fold change (log2FC). Genes were considered differentially expressed at *p* < 0.05 and considering −1.5 > FC > 1.5. Comparative analyses were performed for WES− sows vs. CTR− sows and both Sp-supplemented groups vs their respective non-supplemented groups. Differentially expressed genes were subjected to functional analyses using the Ingenuity Pathway Analysis (IPA) software (version 76765844, Qiagen, Hilden, Germany, www.qiagen.com/ingenuity (accessed on 10 January 2021)). The overlap *p*-value was calculated using Fisher′s exact test, and the z-score provides a prediction of the directional effect.

### 2.7. Validation of Transcriptome Data via qPCR

Reverse transcription was performed using 1 µg of hepatic RNA of all experimental animals with the GoScript Reverse Transcription System (Promega, Madison, WI, USA) according to the manufacturer′s instructions. Quantitative real-time polymerase chain reaction was performed using the KAPA Sybr Fast Mix (Kapa Biosystems, Wilmington, NC, USA) according to the manufacturer′s instructions on a CFX384 Real-Time PCR Detection System (Bio-Rad, Munich, Germany) with a 2-step amplification program (1 s at 95 °C, 30 s at 60 °C, 39 cycles). Primers (Table 2) were ordered at Microsynth (Balgach, Switzerland). Six reference genes were tested (ubiquitin B (UBB), actin B (ACTB), H3.3 histone A (H3F3A), hypoxanthine phosphoribosyltransferase 1 (HPRT1), tyrosine 3-monooxygenase/tryptophan 5-monooxygenase activation protein zeta (YWHAZ), succinate dehydrogenase complex flavoprotein subunit A (SDHA)). The RefFinder web tool was used to determine the most stable combination of reference genes for normalization of target gene expression [29]. The three most stable reference genes (H3F3A, YWHAZ, and UBB) were selected using the recommended comprehensive ranking, and the geometrical mean of their Cqs was used to calculate the ΔCq value of each target gene. Results are shown as log2 fold change, where the fold change was calculated as 2^−ΔΔCq^.

### 2.8. Compositional Analyses of Diets, Liver, Skeletal Muscle, and Feces

In all samples analyzed, compositional analyses were performed with similar methods. For analyses of dry matter (DM) and total ash (method 942.05; AOAC International, 1995), a thermo-gravimetric device (TGA 701, Leco Corporation, St. Joseph, MI, USA) was used. Using a C-N analyzer (Leco-CN 2000, Leco Corporation), nitrogen (N) content was determined (method 968.06; AOAC International, 1995), and crude protein was calculated as 6.25 × N. Ether extract was assessed by Soxhlet extraction using petroleum ether as a solvent (B-811, Büchi, Flawil, Switzerland; AOAC index no. 963.15). The gross energy was measured using a bomb calorimeter (Calorimeter C7000, IKA-Werke GmbH & Co. KG, Staufen, Germany). The cholesterol was extracted as described by Madzlan [30] and measured with a gas chromatograph (HP 6890, Agilent) equipped with a J&W Ultra1 column (25 m × 0.32 mm, Agilent Technologies). Total sugars (glucose, fructose, sucrose, lactose, maltose, xylose, and galactose) and starch content in the diets were analyzed by Eurofins Scientific AG (Schönenwerd, Switzerland) using, respectively, an HPAEC-PAD technique and an enzymatic assay (ISO 15914).

Hepatic total lipids were extracted as previously described by Lan et al. [31]. Briefly, 50 mg of frozen liver tissue was weighed, placed in ice-cold isopropanol, homogenized using a MagNA Lyser Instrument (Roche) at 7000× *g* for 30 s, and incubated for 10 min at room temperature while shaking. The samples were centrifuged at 1107× *g* for 10 min. Triglyceride and total cholesterol contents were analyzed in the supernatant using enzymatic reaction assays (Fluitest^®^ TG and Fluitest^®^ Chol, both from Analyticon, Lichtenfels, Germany) according to the manufacturer′s instructions. Glycogen content was measured in both liver and muscle using the Glycogen Assay Kit (Fluorimetric) (Cell BioLabs, San Diego, CA, USA) following the manufacturer′s instructions.

### 2.9. Histological Analyses of Liver, Skeletal Muscle, and Visceral Adipose Tissue

Histological analyses were performed at the Laboratory for Animal Model Pathology (University of Zurich). Samples were trimmed and embedded in paraffin wax. Consecutive sections (3–5 µm) were prepared, mounted on glass slides, and stained in hematoxylin and eosin (Sigma-Aldrich), according to the manufacturer′s instructions. Adipocyte area was measured from microphotographs (Zeiss Axioskop 2, Zeiss, Jena, Germany) using the software ImageJ (v1.53e, National Institute of Health, Bethesda, Annapolis, MD, USA).

### 2.10. Blood Collection

During physical restraint with a snare, blood samples were collected at the end of the pre-gestation period (T1) and about 1 month before the expected parturition (T2) after 12 h of overnight fasting via jugular venipuncture into serum and plasma tubes (Z; clot activator, and K3-EDTA, S-Monovette, Sarstedt AG&Co. KG, Nuembrecht, Germany). At slaughter (T3), blood was collected during exsanguination into serum and plasma tubes. Tubes were centrifuged at 1500× *g* for 10 min at 4 °C. Serum and plasma supernatants were collected and stored at −80 °C until analyses.

### 2.11. Serum and Plasma Assays

Triglyceride and total cholesterol concentrations were determined in plasma by enzymatic reagent kits following the manufacturer’s instructions (Fluitest^®^ TG and Fluitest^®^ Chol). Glucose was measured in serum samples using Multi-purpose kits—Glucose GOD FS (Dyasys, Diagnostic Systems GmbH, Holzheim, Germany). Hepatic biomarkers were analyzed in serum samples. In particular, alanine transaminase (ALT) and aspartate transaminase (AST) were analyzed using ALAT (GPT) FS and ASAT (GOT) FS (both from Dyasys), respectively. Concentrations of bilirubin, creatinine, gamma-glutamyl transferase (GGT), and glutamate dehydrogenase (GLDH) were analyzed using Bilirubin Auto Total FS (Dyasis), Creatinine FS (Dyasys), Gamma-GT FS (Szasz mod. /IFCC stand.) (Dyasys), and GLDH FS DGKC (Dyasys). Concentrations of glucose, creatinine, AST, GGT, GLDH, bilirubin, and ALT were measured using an automated analyzer (Cobas Miras Plus instrument, Roche Diagnostics AG, Rotkreuz, Switzerland). The insulin-like growth factor 1 (IGF1) was measured using a radioimmunoassay as described in detail by Vicari et al. [32]. Plasma insulin levels were measured using an ELISA kit (Mercodia, Uppsala, Sweden). The HOMA-IR index (Homeostasis model assessment of insulin resistance) evaluating the insulin resistance [33] was calculated from values obtained for fasting glucose (mmol/mL) and insulin (μIU/mL) using the formula: (Insulin × Glucose)/22.5. Total antioxidant capacity (TAC) and the ferric reducing antioxidant power (FRAP) were measured in serum using the OxiSelect Total Antioxidant Capacity Assay kit (Cell BioLab, San Diego, CA, USA) and the OxiSelect Ferric Reducing Antioxidant Power Assay kit (Cell BioLabs) according to the manufacturer′s instructions.

### 2.12. Fatty Acid Analyses in Diets and Tissues

The FA in diets, liver, and abdominal muscle were analyzed as described in detail by Wolf et al. [34]. Briefly, from diets and tissues, FA were extracted with hexane:isopropanol in a ratio of 3:2 (*v/v*) using an accelerated solvent extractor (ASE 200, Dionex Corporation, Sunnyvale, CA, USA). The C11:0 (Fluka Chemie, Buchs, Switzerland) was used as internal standard. Methylation to FA methyl esters (FAME) was performed according to IUPAC (method 2.301). A gas chromatograph (HP 6890, Agilent Technologies) equipped with a CP7421 column (200 m × 0.25 mm, Varian, Lake Forest, IL, USA) was used to separate the FAME. Sunflower oil was used to produce a standard curve. Peak identification was further confirmed using chromatograms from Collomb and Buehler [35].

### 2.13. Statistical Analyses

Due to the experimental setup, sample sizes differed between the different experimental periods. The pre-gestation period with two experimental diets resulted in n = 12 per group. After forming the four experimental subgroups, three animals had to be excluded after the pre-gestation period because of not becoming pregnant, and one animal (WES+ group) had to be excluded because of miscarriage, resulting in group sizes: CTR− n = 5, CTR+ n = 6, WES− n = 5, and WES+ n = 4 for the gestation period. During the lactation period, one CTR+ sow experienced excessive weight loss and had to be euthanized, thus CTR+ was reduced to n = 5 in the lactation and post-lactation period as well as at slaughter. A CTR− animal was excluded due to agalactia from analyses during the lactation period (thus CTR− n = 4 in the lactation period). One of the non-pregnant WES+ animals was re-introduced into the experiment for the post-lactation period and slaughter (thus n = 5 for all experimental groups). All statistical analyses were performed using R [36] (version 4.1.2, R Core Team, Vienna, Austria). The packages lme4 and lmerTest were used to create linear mixed effect models, considering diet, spirulina supplementation, and their nested interaction as fixed effects. Animal, year of experiment, and mother were included as random effects. Normal distribution was confirmed by graphically inspecting the residuals and by using the Shapiro-Wilk test. When the fixed effects met a statistical significance, the normalized datasets were submitted to pairwise comparisons among least squares means using the emmeans package (v1.6.0, Russell V. Length, Iowa, USA). Differences were considered significant at *p* < 0.05 and a trend at 0.05 < *p* < 0.10.

## 3. Results

### 3.1. Dietary Intake and Growth Performance

Feed intake did not significantly differ during any of the experimental periods between WES and CTR animals (Figure 2A). Therefore, WES animals had a significantly higher energy intake during the pre-gestation, gestation, and lactation periods (Figure 2B). The fat and sugar intakes in WES animals were significantly higher compared to CTR animals by about 5–6-fold and 6–7-fold, respectively, during all experimental periods (Figure 2C,D). In turn, the CTR animals had a significantly higher starch intake of about 5–6-fold during all experimental periods (Figure 2E). The supplementation of Sp did not significantly affect the feed intake and thus also did not affect the intake of energy, fat, sugar, and starch (Figure 2A–E).

With the exception of a higher BW in CTR compared to WES sows shortly before farrowing (279 vs. 269 ± 3.4 kg), the animals′ BW did not significantly differ between the dietary groups during the entire experiment (Figure 2F). Both absolute BW gain (data not shown) and average daily gain (ADG) were not significantly affected during any of the four experimental periods (Figure 2G). The Sp supplementation did not affect animal growth during any of the experimental periods (Figure 2F,G). The subcutaneous adipose tissue (SCAT) thickness did not significantly differ at any time point (Figure 2H). The absolute fat retention was about seven-fold higher in WES compared to CTR animals (0.30 vs. 0.04 ± 0.031 kg), while it was not significantly affected by Sp supplementation (Figure 2J).

### 3.2. Differential Impact of WES Diet on Hepatic and Skeletal Muscle Transcriptome

RNA sequencing of liver samples from WES− and CTR− sows revealed 446 differentially abundant transcripts (DATs), among them 183 with lower and 263 with higher abundance in the WES− compared to the CTR− group (Figure 3A). Biological functions related to glucose metabolism (glucose metabolism disorders, quantity of carbohydrates) and weight gain were predicted to be inhibited, while pathways related to lipid metabolism (FA metabolism, lipid oxidation, and quantity of protein-lipid complex in blood) were predicted to be activated (Figure 3B). Moreover, predicted activation of pathways related to hepatic inflammation and oxidative stress (inflammation of organ, chronic inflammatory disorder, quantity of reactive oxygen species) and liver damage (liver lesion, hepatic steatosis) was observed in WES− compared to CTR− sows. Particularly, analyses on regulator effects predicted inhibition of IGF1 signaling, activating the necrotic pathway (Figure 3C). In muscle samples, 729 DATs (116 with lower and 613 with higher abundance in the WES− group) were revealed (Figure 3D). In contrast to the liver, biological functions related to glucose metabolism were predicted to be activated in WES− compared to CTR− sows in the skeletal muscle (diabetes mellitus, metabolism of polysaccharides and carbohydrates), while differences in lipid metabolism were only indicated by a predicted activation of FA uptake (Figure 3E). Biological functions related to immune cell functionality (proliferation and cell death of immune cells, leukocyte migration, immune response of cells) were predicted to be activated in the muscle of WES− compared to CTR−animals. Upstream analyses showed predicted activation of both pro- and anti-inflammatory cytokines, such as tumor necrosis factor (TNF), interleukin 6 (IL-6), and interleukin 10 (IL-10), as well as a predicted inhibition of the aryl hydrocarbon receptor (AHR), thus suggesting a regulatory effect of the WES diet on inflammatory pathways also in the muscle tissue (Figure 3F).

### 3.3. Changes in Gene Expression Influenced by Spirulina

Spirulina supplementation altered the hepatic gene expression profile both in sows on the control diet (442 DATs; 188 with higher and 254 with lower abundance in CTR+ vs. CTR− samples) and on WD (524 DATs; 249 with higher and 275 with lower abundance in WES+ vs. WES− samples). Notably, only 15 and 25 transcripts were commonly up- and downregulated by Sp supplementation in both diet groups (Figure 4A). Dietary Sp resulted in some similarly regulated pathways in both diet groups (Figure 4B). Indeed, Sp supplementation may have inhibited biological functions such as metabolism of carbohydrates and FA, concentrations of cholesterol, D-glucose, and triacylglycerol, insulin resistance, inflammatory response, and apoptosis in livers of both diet groups (Figure 4B). On the other hand, Sp supplementation may have induced a diet-dependent regulation of the necrotic pathway that was predicted to be inhibited in CTR+ compared to CTR− but activated in WES+ compared to WES− sows. Likewise, biological functions related to the quantity of carbohydrates, insulin, and steroids in blood as well as size of body and lipid synthesis were predicted to be differentially regulated, showing activation in livers of CTR+ compared to CTR− but inhibition in WES+ compared to WES− animals.

In the muscle samples from sows fed the control diet, Sp supplementation altered the transcript abundance of 531 genes (397 up- and 134 down-regulated in CTR+ vs. CTR- samples). Muscle samples from WES+ sows revealed 262 DATs (172 up- and 90 down-regulated compared with WES− samples). The two diet groups had only 10 DATs in common—seven with increased and three with decreased abundances after Sp supplementation (Figure 4C). Among the differentially regulated biological functions, only weight gain was predicted to be similarly regulated in the skeletal muscle of both diet groups, being inhibited in Sp-supplemented compared to non-supplemented animals (Figure 4D). Biological functions such as apoptosis, cell movement of leukocytes, cellular infiltration by leukocytes, diabetes mellitus, and growth of connective tissue were predicted activated in CTR+ compared to CTR− sows but inhibited in WES+ compared to WES− sows. In contrast, inflammation was predicted to be inhibited in CTR+ compared to CTR− sows and activated in WES+ compared to WES− sows.

### 3.4. Validation of RNAseq by qPCR

Selected DATs related to lipid and glucose metabolism, inflammation, and oxidative stress identified by RNAseq analysis were validated using qPCR. Expression of hepatic cytochrome P450 family 1 subfamily A member 2 (CYP1A2) and prostaglandin D2 synthase (PTDGS) was confirmed to be significantly up-regulated in WES groups compared to CTR groups (Figure 5A,B), while expression of glutathione S-transferase alpha 4 (GSTA4) was confirmed to be significantly down-regulated in WES vs. CTR sows (Figure 5C). In livers of Sp-supplemented groups, a significantly lower expression of GSTA4 (Figure 5C) and a significantly higher expression of the C-reactive protein (CRP) were observed compared to non-supplemented groups (Figure 5F). According to the RNAseq data, the hepatic expression of IGF1 did not differ due to diet or Sp supplementation, which was confirmed by qPCR (Figure 5H).

### 3.5. Organ Proportions, Composition, and Histology

The proportions and composition of liver, skeletal muscle, and visceral adipose tissue are presented in Table 3. The Sp-supplemented compared to non-supplemented sows had a significantly lower relative liver weight (10.0 vs. 11.6 ± 0.41 g/kg BW). The WES sows had a significantly higher cholesterol (55.8 vs. 53.6 ± 0.83 mg/100 g) and a significantly lower glycogen (7.0 vs. 13.3 ± 1.29 μM/g) content in muscle tissue. The Sp supplementation significantly increased the intramuscular cholesterol concentration compared to non-supplemented animals (55.9 vs. 53.5 ± 0.84 mg/100 g). No difference was observed concerning the relative VAT weight. A significant interaction of experimental diet and Sp supplementation was found for adipocytes with an area of 15–25 × 103 µm^2^. Histological examination did not reveal any visible structural differences in liver, skeletal muscle, and VAT (Appendix A).

### 3.6. Blood Biochemical Biomarkers

Spirulina supplementation resulted in a significantly lower serum glucose concentration (4.82 vs. 5.25 ± 0.179 mmol/L) during late gestation compared to non-supplemented animals (Figure 6A). Plasma IGF1 concentrations were significantly lower in WES compared to CTR sows (Figure 6D) at all three time points. Plasma total cholesterol concentrations were significantly higher in WES compared to CTR sows at slaughter (123 vs. 116 ± 2.3 mg/dL) but at none of the other time points (Figure 6B).

At the end of the pre-feeding period, serum bilirubin and ALT were significantly higher in WES compared to CTR sows (Figure 7A,B). Also during late gestation, significantly higher values were observed for serum bilirubin and ALT (Figure 7A,B) in WES compared to CTR animals (82 vs. 44 ± 5.2 U/L and 5.0 vs. 2.4 ± 0.43 µmol/L, respectively). For ALT and GLDH, a significant diet × Sp supplementation interaction was observed during late gestation (Figure 7B,D), with lower values detected in CTR+ compared to CTR− and higher values detected in WES+ compared to WES− animals.

### 3.7. Fatty Acid Composition of Liver and Skeletal Muscle

Diet and Sp supplementation influenced the FA profile of liver and skeletal muscle at slaughter to different extents (Table 4). In liver tissue, the proportions of total SFA were significantly higher in WES compared to CTR sows. Significantly lower total n–6 FA were observed in the livers of WES compared to CTR animals without significantly affecting the n–6/n–3 FA ratio. The proportions of LA, C20:0, C20:1 n–9, C20:2 n–6, and C22:4 n–6 were significantly lower, while the proportions of DGLA and EPA were significantly higher in WES compared to CTR sows (Appendix A). Supplementation with Sp hardly modified the liver FA profile with significantly lower proportions of C17:0 and DGLA (Appendix A).

In muscle tissue, diet resulted in higher C17:0 and C18:0 proportions in WES compared to CTR sows (Appendix A). The proportions of C20:1 n–9 and C20:2 n–6 (Supplementary Table 1) as well as the n–3/n–6 FA ratio (Table 4) were significantly lower in WES compared to CTR sows′ muscle. The Sp supplementation resulted in significantly lower intramuscular C17:0 and C18:0 proportions (Appendix A). A significant interaction of diet and Sp supplementation was observed for C17:0, with WES− sows having higher proportions than all other groups (Appendix A). In addition, for intramuscular ALA proportions, a significant interaction of diet and Sp supplementation was observed. The ALA proportion was higher in CTR+ than in CTR− animals, while it was lower in WES+ than in WES− animals.

## 4. Discussion

Feeding a WD to domestic pigs before and during gestation as well as during and after lactation was not sufficient to induce an obese phenotype. Still, some metabolic dysfunctions occurred, as indicated by the significantly lower IGF1 concentrations as well as an activation of diabetes mellitus–related pathways in muscle tissue of WES compared to CTR animals. Moreover, serum hepatic biomarkers before and during gestation as well as hepatic cholesterol accumulation and gene expression at slaughter might indicate the onset of non-alcoholic steatohepatitis (NASH) in WES sows. The supplementation of Sp reduced plasma insulin levels in WES+ compared to WES− sows at slaughter to similarly low levels as observed in CTR− and CTR+ sows. However, Sp was not able to counteract the onset of NASH observed in WES sows.

### 4.1. Effects of a Western Diet on the Metabolic Phenotype of Female Domestic Pigs

#### 4.1.1. Western Diet Intake Induced Regulatory Processes to Maintain Body Weight and Visceral Adipose Tissue Proportion

Despite the higher intake of gross energy, fat, and sugars in WES compared to CTR sows throughout the experiment, the BW, relative VAT weight, and SCAT thickness did not differ between WES and CTR animals at any time point. This contrasts with previous literature in other porcine models [37,38,39,40,41,42,43], where high energy and/or high-fat, high-sugar diets increased the BW by at least 60% and the VAT mass by twofold compared to control groups. However, most of these studies used minipigs [37,38,39,40,41,43], which are more susceptible to BW gain and visceral adiposity in response to an obesogenic diet than domestic pigs [18].

The observed lack of BW differences in the present study might be explained by the IPA-predicted activation of lipid oxidation in the liver and FA uptake in the muscle of WES− compared to CTR− sows. Switching between glucose and FA as an energy source indicates metabolic flexibility, which allows an organism to successfully respond to changes in metabolic and energy demand [44]. The lack of both tissue triglyceride accumulation as well as increased BW of WES compared to CTR animals points towards an increased FA utilization in liver and skeletal muscle, which may be explained by a higher metabolic flexibility of the WES animals.

Even though the pigs in the present study did not develop an obese phenotype, it must be considered that the “lean metabolic syndrome” (LMS) has recently gained increasing attention in human medicine [45]. The LMS patients have a normal BW and VAT proportion but still show two or more characteristics of the metabolic syndrome [46,47]. Therefore, the WES pigs used in the present study might be a particularly suitable model for the human LMS as indicated by metabolic characteristics that are discussed in the following sections.

#### 4.1.2. Liver Functionality Was Adversely Affected by the Western Diet

Elevated systemic transaminase (ALT, AST) and bilirubin levels as indicators for hepatic damage have been reported in rodents and human patients with obesity and fatty liver [48,49,50,51]. Serum bilirubin and ALT were about twofold higher in WES compared to CTR animals after the pre-feeding period at AI as well as one month before farrowing, indicating an impaired liver functionality in the WES compared to the CTR sows during gestation. Although transcriptome analysis of liver tissue predicted activation of pathways related to liver stress and damage in WES− compared to CTR− sows, these systemic markers did not significantly differ in our sows following the lactation period at slaughter. We suggest that catabolic processes occurring during lactation might have normalized systemic hepatic biomarkers at slaughter. Similarly, in women with a history of gestational diabetes, breastfeeding reset the maternal metabolism in the early post-partum period [52].

Enhanced inflammation and oxidative stress, as observed in obesity and LMS [53,54], were predicted by pathway analysis in livers of WES− compared to CTR− animals. Since the serum TAC and FRAP in the present study did not significantly differ between diet groups, we suggest that increased ROS levels and associated gene expression, as predicted by IPA of liver samples and confirmed by qPCR, might have been limited to liver tissue without affecting the systemic antioxidant status. The family of glutathione S-transferase (GST) enzymes is able to protect against ROS-induced damage by regulating tissue regeneration pathways in the liver [55,56,57]. A reduced hepatic expression of the GSTA1 isoform is considered a marker of hepatic injury [56]. Expression of both GSTA1 and GSTA4 was lower in the livers of WES compared to CTR sows, thus further confirming a certain extent of liver damage and impaired hepatic regenerative capacity in WES sows.

Oxidative stress and inflammation in liver tissue are caused, e.g., by a hepatic lipid accumulation [58]. Transcriptome activation of pathways related to hepatic steatosis was observed in WES− compared to CTR− sows. The higher proportions of hepatic SFA in WES compared to CTR sows reflected the higher SFA proportion of the WES feed. Continuous exposure to high proportions of dietary SFA has been shown to promote the accumulation of SFA in liver tissue and thus the development of fatty liver disorders by inducing pro-inflammatory pathways and stimulating apoptosis due to endoplasmic reticulum and oxidative stress [59]. The significantly higher hepatic SFA proportions in WES compared to CTR animals at slaughter thus further support a prevalent impaired liver functionality in WES sows. Similar to obese humans, a frequent occurrence of NASH or nonalcoholic fatty liver diseases has been observed in human LMS patients [60,61].

In line with this, the most strongly activated biological function identified by pathway analysis indicated the occurrence of necrotic processes in the liver of WES− compared to CTR− sows. This was also predicted to be induced by reduced IGF1 signaling. The IGF1 is a growth hormone that has been shown to exert hepatoprotective effects by promoting tissue regeneration in the case of injury [62]. In line with the predicted reduction of IGF1 signaling, lower plasma IGF1 concentrations were indeed observed in WES compared to CTR animals throughout the entire experimental period including slaughter. Reduced IGF1 abundance and thus signaling suggest an impaired regenerative capacity of the liver in WES sows, also matching the observed (temporary) increase in serum bilirubin and transaminases.

#### 4.1.3. The Impaired Glucose Metabolism Induced by the Western Diet Was Likely Mediated by Metabolic Changes Predominantly in the Skeletal Muscle

Diabetes mellitus related pathways were activated in the muscle of WES− compared to CTR− sows. Interestingly, pathways for glucose metabolism disorder were instead inhibited in both liver and muscle. This may indicate that the increased intake of fat and sugars indeed activated diabetes mellitus–related pathways in the muscle but that in both muscle and liver, processes counteracting these negative WD-induced effects were activated. Previous studies in minipigs have similarly failed to induce diabetes mellitus through feeding [63]. The main reason why pigs do not easily develop diabetes lies in the evolutionary history of wild pigs, which had to store energy during the summer to be later used in the winter when food availability was lower, thus building a pancreatic resistance to diabetogenic diets [18]. Still, at all time points, WES compared to CTR sows had lower plasma IGF1 concentrations as observed in patients with a long history of diabetes [64]. A reduced glycogen content in muscle, as observed in WES compared to CTR sows, was also found in both type 1 and 2 diabetic patients [65]. Taken together, results from the present study point towards a mild diabetogenic-like phenotype that was induced by WD feeding in pigs.

### 4.2. Effects of Spirulina Supplementation on the Metabolic Phenotype of Female Domestic Pigs

#### 4.2.1. Spirulina Supplementation Did Not Affect Body Weight but Potentially Aggravated WD-Induced Liver Damage

In humans, Sp supplementation has been shown to promote weight loss via regulation of food intake and adipokines [66]. In contrast, Sp supplementation did not affect either feed intake or BW of the adult but still growing sows in the present study. This also contrasts with the predicted inhibition of weight gain obtained from gene expression analysis in muscle tissue of Sp-supplemented compared to non-supplemented pigs.

Despite the similar BW and VAT proportion, Sp supplementation led to a diet type–independent significantly decreased relative liver weight compared to non-supplemented animals. Liver weight differences are usually caused by differences in fat or glycogen accumulation or by hepatic degeneration [67]. The similar hepatic lipid and glycogen concentrations in Sp-supplemented and non-supplemented groups contradict the IPA prediction of Sp-mediated higher hepatic lipid and lower hepatic carbohydrate concentration in WES compared to CTR animals. The significant interaction of diet and Sp supplementation for plasma ALT and GLDH might point towards a diet-dependent effect of Sp on liver functionality that even aggravated the adverse effects of the WD. In line with this, pathway analyses showed that Sp supplementation in the CTR+ group inhibited hepatic necrosis, while this pathway was activated in the WES+ group. Therefore, Sp in combination with the WES diet in pigs seems to not exert the hepatoprotective effects previously described in rodent models fed a WD [14,68]. 

#### 4.2.2. Spirulina Supplementation Had Only Small Diet-Dependent Effects on Systemic Metabolic Biomarkers

Inhibition of insulin resistance and activation of glucose metabolic disorders were predicted by hepatic and muscle gene expression, respectively, in Sp-supplemented compared to non-supplemented animals, independent of diet type. However, a significant interaction of diet and Sp supplementation was observed for plasma insulin concentration at slaughter. In line with the gene expression prediction, insulin concentrations of WES+ sows were similarly low to those of the CTR− and CTR+ sows. These findings suggest that Sp supplementation in WD-fed pigs may maintain plasma insulin concentrations at a normal level via a yet unknown mechanism.

Plasma lipids and antioxidant capacity were not affected by Sp supplementation. In previous reports, Sp supplementation has been shown to exert hypolipemic effects in both rodent models of obesity and in obese patients [16,66]. These individuals were, however, neither pregnant nor lactating. Given the extensive metabolic changes during pregnancy and lactation, those study results can therefore not be directly compared to the results of the present study.

#### 4.2.3. Spirulina Supplementation Had Only Minor Effects on the Tissue Fatty Acid Profiles

Although Sp is rich in GLA and LA as well as in other bioactive components potentially affecting FA metabolism, Sp supplementation had very little effect on the FA profiles of liver and muscle in the present study. Importantly, the proportions of GLA were higher in the liver of Sp-supplemented versus non-supplemented animals. Hepatic incorporation of GLA plays a pivotal role in reducing fat deposition by enhancing the activity of enzymes involved in FA oxidation [69]. However, no such effect was observed in the present experiment, as hepatic lipids were not affected in Sp-supplemented compared to non-supplemented animals.

### 4.3. Limitations of the Study

Feeding a WD to adult but still growing domestic sows did not induce a typical obesity phenotype as intended. Therefore, using adult sows as a translational model for human obesity does not seem feasible. However, pigs grow faster before puberty [70]. Starting the WD feeding at an earlier age and/or extending the experimental feeding period to potentially increase the magnitude of the adverse metabolic effects might thus have a considerable impact on body composition, as was shown by Fisher et al. [71], who fed the same proportions of fats and sugar to growing pigs.

The composition of the CTR diet might also have introduced a bias since it was rich in starch, and starch-rich diets were previously demonstrated to be also positively associated with metabolic syndrome [72]. Therefore, the lack of significance in some parameters may be due to the fact that the CTR group also consumed a diet that could potentially lead to similar disturbances and would not be considered particularly healthy in human nutrition.

Importantly, the metabolic changes naturally occurring during gestation and lactation likely interacted with the experimental diets. Despite the absence of an obesity phenotype, the sows presented with some metabolic abnormalities manifesting in the protein level during pre-gestation and gestation. The indications of metabolic abnormalities on the hepatic and muscle transcriptome level at the end of lactation could, however, not be confirmed on the protein level at the same time point. The experimental design limited tissue sample collection to the end of the 5-week lactation period, which is known to cause significant metabolic adaptations via catabolic processes that potentially ameliorated the detrimental effects that were indicated during gestation [52]. Analyses of liver and muscle biopsies during gestation would have provided more reliable insights since the current data do not allow us to exclude with certainty that significant differences in lipid accumulation and thus a fatty liver were present before or during gestation.

The few effects of Sp supplementation on the mitigation of detrimental WD effects might be related to the rather low dosage applied. Previous studies have shown that higher Sp dosages were used to achieve beneficial health effects in pigs [73]. Still, the amount supplemented in the present study corresponded to the amount recommended in human nutrition relative to BW. The lack of mitigating Sp effect might, however, also be due to the absence of an obesity phenotype in our WES sows. Therefore, future studies on the potential beneficial effects of Sp supplementation on obesity-induced metabolic disturbances should be performed in animals showing a pronounced obesity phenotype, as can be observed in humans.

Overall, it has to be mentioned that group sizes in the present study were rather small, and larger studies should be designed to investigate the observed effects in more detail.

## 5. Conclusions

Our hypothesis that (i) feeding a WD to domestic pigs induces an obesity-like metabolic phenotype similar to that observed in humans is only partially confirmed by the study results. The variable effects on metabolic biomarkers observed during the different physiological states confirm the hypothesis that (ii) the effects of the WD differ during pre-gestation, gestation, lactation, and post-lactation periods. The hypothesis that (iii) low-dose supplementation of Sp may attenuate or even prevent the unfavorable phenotype observed in pigs fed a WD was only partially confirmed by the normalized plasma insulin levels in WES+ animals, representing the only counter-regulatory effect observed. Further investigations are of interest to determine if Sp supplementation may prevent more pronounced adverse metabolic effects in animals with more severe diet-induced metabolic impairments and if higher Sp doses might be more effective. In addition, more in-depth analyses, particularly of liver and muscle tissue during gestation and lactation, are warranted to understand the nutritional impact of WD and Sp supplementation during such special metabolic events.

## Figures and Tables

**Figure 1 nutrients-14-03574-f001:**
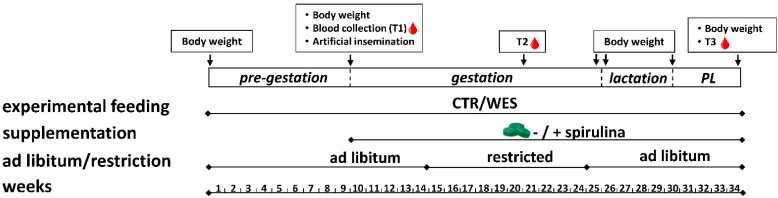
Schematic representation of the feeding experiment performed on female crossbred pigs in two runs (2019 and 2020) from spring to winter (34 weeks in total). Experimental sows were fed a control (CTR) or Western (WES) diet from the pre-gestation period to the post-lactation (PL) period for a total of 34 weeks. At artificial insemination, spirulina supplementation was started for half of both diet groups. Body weight was recorded at the start and end of each experimental period. Blood was collected at three time points (T1, T2, T3).

**Figure 2 nutrients-14-03574-f002:**
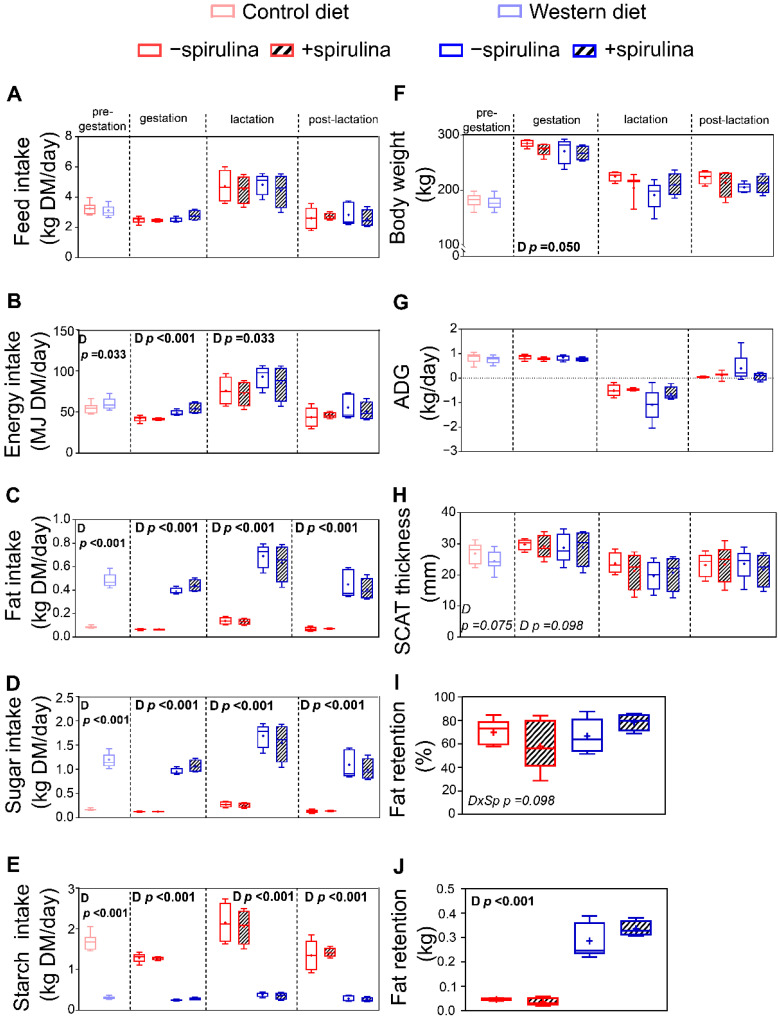
Effects of diet and spirulina supplementation on feed intake and growth performance of the sows during pre-gestation, gestation, lactation, and post-lactation, as well as body fat retention determined one week before expected farrowing. Experimental sows received either a control (CTR) or Western (WES) diet throughout all four experimental periods, while spirulina supplementation was started for half of CTR/WES groups at the end of the pre-gestation period. Differences in (**A**) feed, (**B**) energy, (**C**) fat, (**D**) sugar, and (**E**) starch intake during the four experimental periods; (**F**) body weight (BW) of experimental animals at the end of each period; (**G**) average daily gain (ADG) of experimental animals during each experimental period; (**H**) subcutaneous adipose tissue (SCAT) thickness at the end of each experimental period; (**I**) relative fat retention and (**J**) absolute fat retention one week before expected farrowing calculated from fat intake and fecal fat excretion during 24 h. Sp: spirulina; D: diet. Data are presented in boxplots with Spear style whiskers (min to max).

**Figure 3 nutrients-14-03574-f003:**
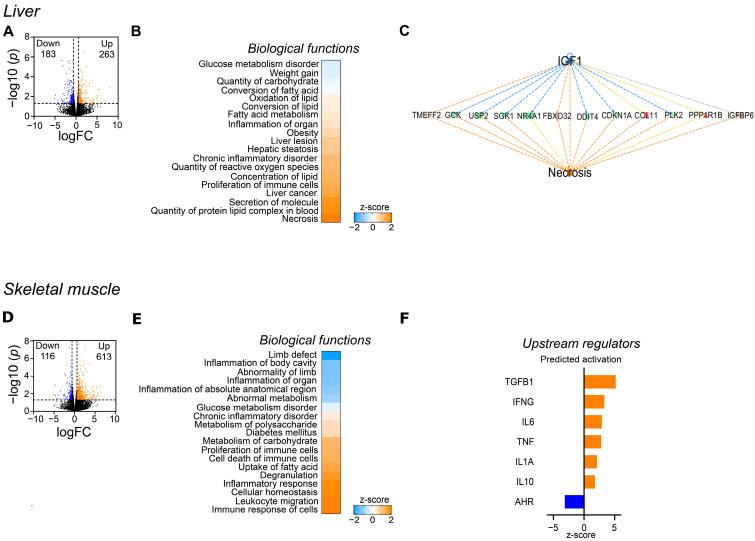
Effects of the WES diet on gene expression of liver and skeletal muscle at slaughter. Only CTR− and WES− animals were included in this analysis. (**A**) Volcano plot representing the RNAseq results in liver in WES− sows compared to CTR− sows. Orange dots indicate the differentially abundant transcripts (DATs) that are significantly up-regulated (cut-off of *p* < 0.05 and −1.5 > fold change > +1.5), while the blue dots represent the down-regulated DATs (cut-off of *p* < 0.05 and −1.5 > fold change > +1.5). (**B**) Heatmap representing the z-score of biological functions significantly affected by WES diet (calculation performed in IPA) in the liver. Blue indicates a negative z-score (inhibition) and orange a positive z-score (activation). (**C**) Regulatory effect of IGF-1 on the hepatic necrosis pathway. (**D**) Volcano plot representing the RNAseq results in skeletal muscle. Orange dots indicate the up-regulated DATs (cut-off of *p* < 0.05 and −1.5 > fold change > +1.5), while the blue dots represent the down-regulated DATs (cut-off of *p* < 0.05 and −1.5 > fold change > +1.5). (**E**) Heatmap representing the z-score of biological functions significantly affected by WES diet (calculation performed in IPA) in muscle. (**F**) Predicted activation of upstream regulators in skeletal muscle of WES− sows.

**Figure 4 nutrients-14-03574-f004:**
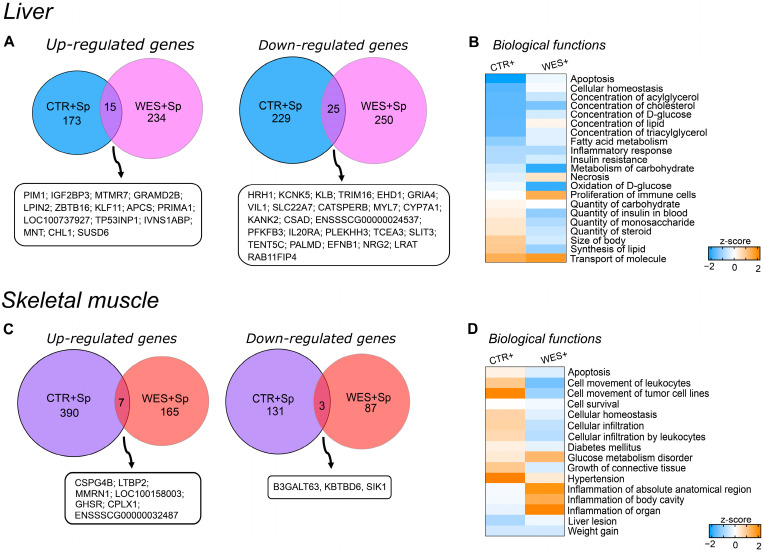
Effects of Sp supplementation on gene expression of liver and muscle at slaughter. (**A**) Venn diagram showing the intersection of differentially abundant transcripts in CTR+ and WES+ livers. Venn diagrams were designed using http://bioinformatics.psb.ugent.be/webtools/Venn/ (accessed on 1 September 2021). (**B**) Heatmap representing the z-score of biological functions significantly affected in both CTR+ and WES+ groups (calculation performed in IPA) in the liver. Blue indicates a negative z-score (inhibition) and orange a positive z-score (activation). (**C**) Venn diagram showing the intersection of differentially abundant transcripts in CTR+ and WES+ muscle. (**D**) Heatmap representing the z-score of biological functions significantly regulated in CTR+ and WES+ groups (calculation performed in IPA) in muscle.

**Figure 5 nutrients-14-03574-f005:**
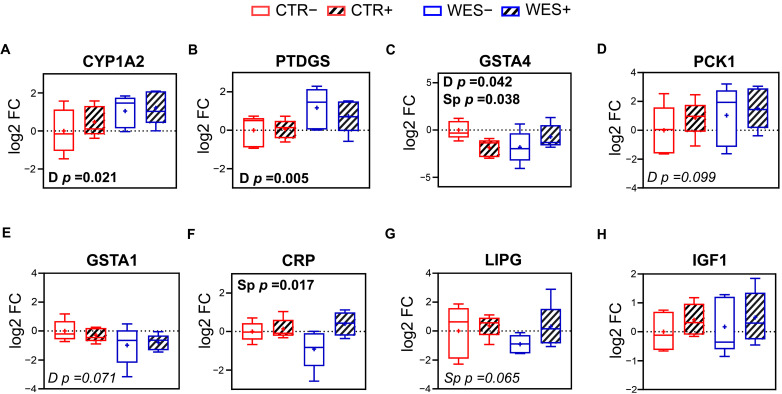
Validation of RNAseq results of selected differentially expressed genes in liver via qRT-PCR. Hepatic levels of expression of (**A**) cytochrome P450 family 1 subfamily A member 2 (*CYP1A2*), (**B**) prostaglandin D2 synthase (*PTDGS*), (**C**) glutathione S-transferase alpha 4 (*GSTA4*), (**D**) phosphoenolpyruvate carboxykinase 1 (*PCK1*), (**E**) glutathione S-transferase alpha 1 (*GSTA1*), (**F**) C-reactive protein (*CRP*), (**G**) lipase G (*LIPG*), and (**H**) insulin-like growth factor 1 (*IGF1*). CTR: control diet; WES: Western diet; D: diet; Sp: spirulina. Data are presented in boxplots with Spear style whiskers (min to max).

**Figure 6 nutrients-14-03574-f006:**
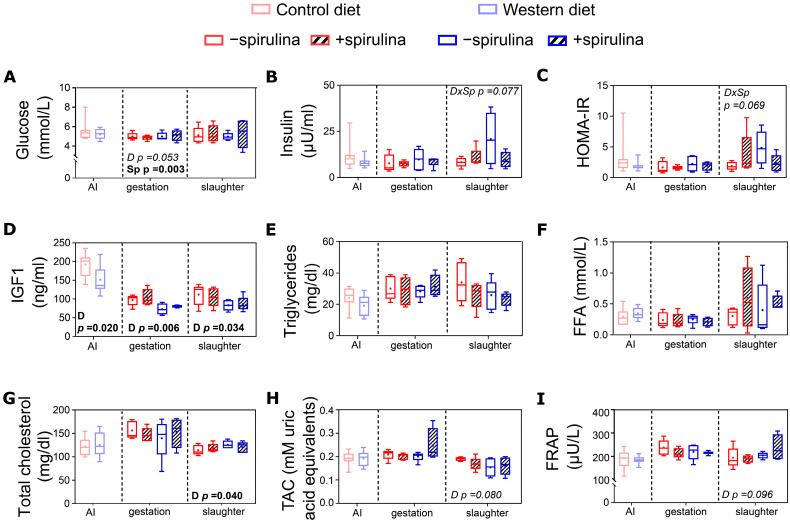
Systemic metabolic and oxidative stress markers of sows at artificial insemination (AI), 30 days before calculated farrowing (gestation), and 5 weeks post-lactation (slaughter). Concentrations of (**A**) serum glucose, (**B**) plasma insulin, (**C**) calculated index for insulin resistance (HOMA-IR), concentrations of (**D**) plasma IGF1, (**E**) plasma triglycerides, (**F)** serum free fatty acids (FFA); (**G**) plasma total cholesterol, and (**H**) serum total antioxidant capacity (TAC) as well as (**I**) serum ferric reducing antioxidant power (FRAP). CTR: control diet; WES: Western diet; D: diet; Sp: spirulina. Data are presented in boxplots with Spear style whiskers (min to max).

**Figure 7 nutrients-14-03574-f007:**
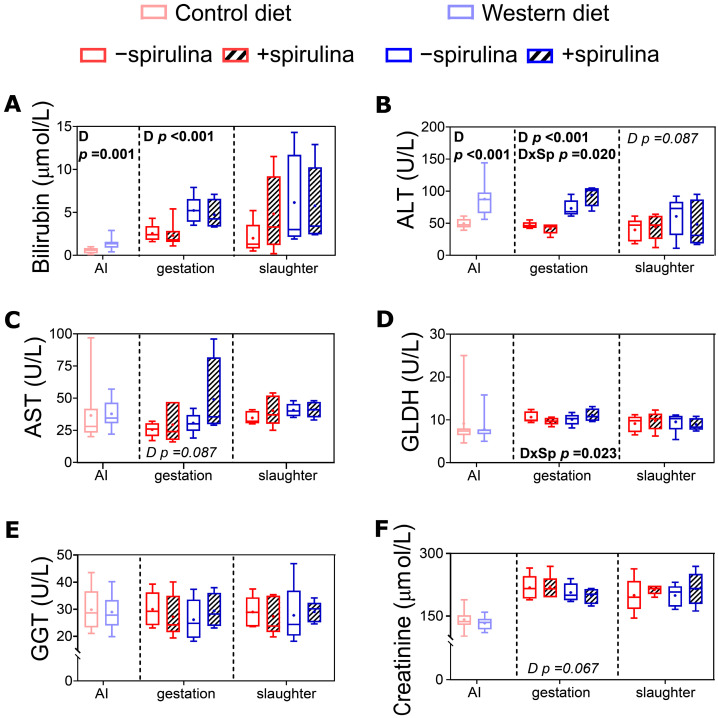
Biomarkers of organ functionality in serum of sows at artificial insemination (AI), 30 days before calculated farrowing (gestation), and 5 weeks post-lactation (slaughter). (**A**) Concentration of serum bilirubin, activities of (**B**) serum alanine transaminase (ALT), (**C**) serum aspartate transaminase (AST), (**D**) serum glutamate dehydrogenase (GLDH)^1^, (**E**) serum gamma-glutamyl transferase (GGT), and (**F**) concentration of serum creatinine. CTR: control diet; WES: Western diet; D: diet; Sp: spirulina. Data are presented in boxplots with Spear style whiskers (min to max). Pairwise comparison showed no significant difference.

**Table 1 nutrients-14-03574-t001:** Ingredients and composition of experimental diets.

	Gestation Diet		Lactation Diet		
	CTR_G_	WES_G_	CTR_L_	WES_L_	Spirulina
Ingredients (g/kg as fed)					
Hydrogenated palm oil	-	150	-	150	-
Saccharose	-	200	-	200	-
_D_-Fructose	-	150	-	150	-
Cholesterol	-	2.0	-	2.0	-
Corn germ	590	298	488	200	-
Wheat	300	-	300	-	-
Soybean meal	45.0	130	145	217	-
Lignocellulose ^†^	26.6	29.0	26.6	29.0	-
Monocalcium phosphate	13.0	20.0	11.2	20.0	-
Calcium carbonate	11.0	6.00	11.6	6.00	-
NaCl	6.2	6.0	6.2	6.0	-
Vitamin and mineral premix ^‡^	5.0	5.0	5.0	5.0	-
_L_-Lysine	2.3	2.5	3.6	5.4	-
_DL_-Methionine	-	1.0	0.7	1.9	-
_DL_-Tryptophan	0.2	0.4	0.5	1.0	-
_L_-Threonine	0.7	1.3	1.8	3.4	-
Valine	-	-	-	2.2	-
Chemical composition (g/kg dry matter (DM) if not stated otherwise)					
DM (g/kg fresh matter)	893	936	898	934	935
Total ash	47.8	47.1	54.9	54.3	90.2
Crude protein	110	93	150	162	638
Ether extract	26	158	29	143	47
Starch	462	93	408	74	51
Total sugars	46	359	52	326	<0.5
Crude fiber	6.7	8.7	6.2	8.1	n.a.
Cholesterol	0.03	0.48	0.02	0.45	n.a.
Gross energy (MJ/kg DM)	13.4	17.2	13.0	16.8	n.a.
Fatty acids (FA) composition (g/100 g total FA) ^§^					
C12:0	0.14	0.64	0.30	0.08	0.03
C14:0	0.15	0.91	0.12	0.11	0.17
C16:0	15.9	30.7	12.3	5.9	42.9
*iso* C16:0	0.06	0.00	0.06	0.00	1.93
C16:1 n–7	0.15	0.01	0.12	0.03	5.44
C16:1*x*	0.34	0.00	0.04	0.00	0.05
C17:1	0.03	0.00	0.04	0.03	0.33
C18:0	4.4	56.4	14.8	81.4	3.4
C18:1 *cis*-9	26.0	3.3	21.9	2.8	1.6
C18:1 *cis*-11	0.88	0.11	0.67	0.12	0.73
C18:2 n–6 (LA)	47.6	5.0	45.1	5.0	16.5
C18:3 n–3 (ALA)	1.93	0.34	2.28	0.68	0.43
C18:3 n–6 (GLA)	0.0	0.0	0.0	0.0	23.2
C20:0	0.49	0.89	0.64	2.00	0.09
C20:1 n–9	0.42	0.04	0.34	0.06	0.12
C20:2 n–6	0.11	0.01	0.05	0.00	0.30
C20:3 n–6	0.00	0.00	0.00	0.00	0.38
C22:0	0.22	0.35	0.35	1.15	0.00
∑ Saturated FA	22.0	90.3	29.1	91.1	49.2
∑ Monounsaturated FA	28.0	4.3	23.3	3.1	8.9
∑ Polyunsaturated FA	50.0	5.40	47.6	5.75	41.9
∑n–6	47.7	5.0	45.2	5.0	40.4
∑n–3	1.9	0.3	2.3	0.7	0.4
n–6/n–3	25.1	14.7	19.6	7.4	101

CTR: Control; FA: fatty acids; WES: Western. n.a.: not analyzed. ^†^ Jeluvet Lignocellulose (Jelu-Werk J. Ehrler GmbH & Co. KG, Rosenberg, Germany). ^‡^ Naco Premix for breeding pigs (Vital AG, Oberentfelden, Switzerland), contains (per kg): lysine, 0.30 g; methionine, 0.10 g; Ca, 174 g; Fe (iron sulphate monohydrate E1), 16 g; native Mg, 8 g; Cu (copper sulphate pentahydrate E4), 3 g; Se (sodium selenite E 8), 0.05 g; Zn (zinc oxide 3b603), 18 g; iodine (calcium iodate anhydride 3b202), 0.2 g; vitamin D_3_, 0.009 g; vitamin A, 0.72 g; vitamin E, 10 g; mineral oil, 10 g. ^§^ Only fatty acids (FA) with a proportion >0.3 g/100 g total FA in at least one of the feed items are displayed.

**Table 2 nutrients-14-03574-t002:** List of primer sequences.

Gene	Forward Primer (5′→3′)	Reverse Primer (5′→3′)	Amplicon Length	Accession Number
ACTB	GATCTGGCACCACACCTTCT	AGAGACAGCACAGCCTGGAT	174	NM_173979.3
CYP1A2	GCCCAGCCCTACTCTGCAA	CCAGGAGATGGCTGTGGTAA	250	XM_005666124.3
CRP	TGAACACAGGCTCTCACATCC	CAAGCCAGACACTTGAATGCC	70	XM_003355107.4
GSTA1	CAGGACACCCAGGACCAATC	GTCTCAGGTACATTCCGGGAG	202	NM_214389.2
GSTA4	GCTCGGAGTGGACCCAGAAAA	TTCGGGTCTGCACCAACTTC	243	NM_001243379.1
H3F3A	AGGAGGTCTCTATACCATGGCTC	GAGCAATTTCCCGCACCAGA	245	NM_213930.1
HPRT1	TGCTGAGGATTTGGAGAAGG	CAACAGGTCGGCAAAGAACT	154	NM_001034035.2
IGF1	TGGTGGACGCTCTTCAGTTCG	ACAGTACATCTCCAGCCTCCTC	155	NM_214256.1
LIPG	CGAAACTCAGTTCCTCTGCTCT	TGGCTGTTGCATTGAAGCCA	247	XM_013992843.2
PCK1	GGGCATCATCTTCGGAGGG	AGTTGTAGCCGAAGAACGGC	182	XM_005673043.3
PTDGS	AAGAACTACGCCCTGCTCCA	ATGGCCAGGTCCTGAGAGT	231	NM_214228.1
SDHA	GCAGAACCTGATGCTTTGTG	CGTAGGAGAGCGTGTGCTT	185	NM_174178.2
UBB	CATTGTTGGCGGTTTCGCT	TTGACCTGTGAGTGAAGGCA	85	NM_001105309.1
YWHAZ	ATTGGGTCTGGCCCTTAACT	GCGTGCTGTCTTTGTATGACTC	146	XM_021088756.1

**Table 3 nutrients-14-03574-t003:** Liver, skeletal muscle, and visceral adipose tissue composition of sows following 34 weeks of control (CTR) or Western (WES) diet feeding. Spirulina supplementation was started from week nine for half of the CTR/WES groups.

Diet (D)	Control Diet		Western Diet		SEM	Significance		
Spirulina (Sp)	−	+	−	+		D	Sp	D × Sp
Liver								
Proportion (g/kg BW)	11.1	10.1	11.8	10.8	1.10	#	*	n.s.
Glycogen (µM/g)	2.17	1.14	2.10	1.41	0.826	n.s.	n.s.	n.s.
Triglycerides (mg/g)	4.20	4.17	4.13	4.23	1.057	n.s.	n.s.	n.s.
Cholesterol (mg/g)	1.33	1.47	1.85	1.95	0.321	#	n.s.	n.s.
Skeletal muscle								
Intramuscular fat (g/kg DM)	55.7	82.7	56.0	52.3	17.30	n.s.	n.s.	n.s.
Protein (g/kg DM)	18.5	19.0	19.3	17.9	1.19	n.s.	n.s.	n.s.
Cholesterol (mg/100g)	51.1	54.9	55.1	56.7	2.53	*	*	n.s.
Glycogen (µM/g)	13.7	12.8	9.81	4.17	2.650	**	n.s.	n.s.
Visceral adipose tissue								
Proportion (g/kg BW)	12.6	11.6	10.1	12.2	1.86	n.s.	n.s.	n.s.
Adipocyte area (µm^2^ × 10^3^)	16.3	12.2	14.2	16.5	3.23	n.s	n.s.	#
Adipocyte proportion (%)								
<5 (µm^2^ × 10^3^)	5.10	5.86	4.05	6.00	6.300	n.s.	n.s.	n.s.
5–15	41.0	66.8	55.5	39.6	24.68	n.s.	n.s.	n.s.
15–25	52.0	23.4	36.1	49.0	15.76	n.s.	n.s.	*
>25	5.68	2.37	4.37	8.85	8.58	n.s.	n.s.	n.s.

Data are presented as least square mean ± standard error of the mean (SEM). Statistical significances were set at * *p* < 0.05 and ** *p* < 0.01. Trends were defined as # 0.05 < *p* < 0.10. n.s.: not significant.

**Table 4 nutrients-14-03574-t004:** Effects of Western diet and spirulina supplementation on the main FA groups present in liver and skeletal muscle of sows (g/100 g total FA).

Diet (D)		Control Diet		Western Diet			Significance		
Spirulina (Sp)		−	+	−	+	SEM	D	Sp	D × Sp
	Tissue								
∑SFA	Liver	41.6	41.6	43.3	42.6	0.78	*	n.s.	n.s.
	Muscle	41.0	39.9	42.6	40.9	1.10	n.s.	#	n.s.
∑MUFA	Liver	18.8	18.4	18.3	21.8	3.47	n.s.	n.s.	n.s.
	Muscle	46.8	47.3	44.9	47.5	2.14	n.s.	n.s.	n.s.
∑PUFA	Liver	39.7	40.0	38.3	35.6	2.87	#	n.s.	n.s.
	Muscle	12.5	12.7	13.1	10.6	2.42	n.s.	n.s.	n.s.
∑n–3	Liver	3.60	3.72	3.90	3.58	4.500	n.s.	n.s.	n.s.
	Muscle	0.58	0.62	0.78	0.63	0.119	n.s.	n.s.	n.s.
∑n–6	Liver	36.0	36.1	34.3	31.9	2.49	*	n.s.	n.s.
	Muscle	11.7	12.0	12.3	9.8	2.31	n.s.	n.s.	n.s.
n–6/n–3	Liver	10.0	9.7	8.8	8.9	0.01	n.s.	n.s.	n.s.
	Muscle	20.2	19.4	15.8	15.6	0.003	***	n.s.	n.s.

SFA: saturated fatty acids; MUFA: monounsaturated fatty acids; PUFA: polyunsaturated fatty acids. Data are presented as least square mean ± standard error of the mean (SEM). Statistical significances were set at * *p* < 0.05 and *** *p* < 0.001. Trends were defined as # 0.05 < *p* < 0.10. n.s.: not significant.

## Data Availability

The Fastq files from the RNAseq experiment were deposited in NCBI′s Gene Expression Omnibus and are accessible through GEO Series accession number GSE197750. All other data are included within the article and supplementary data. Details can be made available upon reasonable request.

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
