# Peer review of "Crossbred Sows Fed a Western Diet during Pre-Gestation, Gestation, Lactation, and Post-Lactation Periods Develop Signs of Lean Metabolic Syndrome That Are Partially Attenuated by Spirulina Supplementation"

_nutrients, 2022, doi:10.3390/nu14173574_

Round 1

Reviewer 1 Report

Overall, the submission represents interesting work describing the investigation the effect of supplementing spirulina on (top of) a western diet-induced cardiometabolic perturbations in gestating and lactating pigs. The study was conducted with extensive analysis (gene expression, lipid analysis and other related risk factors). The level of the description on methods, results and discussion is recognized and appreciated. We provide some comments for consideration and improvement:

1) Grammar check throughout is required.

2) The introduction could be condensed. However, the discussion is particularly long given the proportional data sets being provided. The discussion could be reduced by 30-50%.

3) Please consider editing Figure 1. The information is far too dense for the reader to easily follow. Make the image larger, reduce the number of abbreviations used and please describe the study design in the legend (do not just leave the reader to follow the image per se; this comment is valid for many of the figure legends).

4) Similarly, please make edits to figure 2. Make larger to help readability and do not use abbreviations wherever possible. For example, expand the group abbreviations at the top of the figure. Please add more detail the experiment periods. The figure should be interpretable as a stand-alone entity.

5) For figure 6, please consider switching this dataset to a table. The reviewer does not see any interpretive advantage to have these types of data in the bar graph condensed format.

6) Please expand the panel sizing in Fig 7 and 8. The significance in these images are difficult to read. There is no need for small scale formatting.

5) Lines 159-163 " One week before expected parturition (day 107 of gestation), the diets of both groups were switched to lactation-type diets (CTRL, WESL) which correspondingly provided higher dietary proportions of fat, sugars, and cholesterol to the WES group". Please reword the sentence and clarify relative to data on Table 1. From what Table 1 has stated, the CTRL has less cholesterol than CTRG, please explain/clarify. Please also include the composition of total fat, total sugars to reflect these changes.

6) Line 215, please add a statement/justification as to why only half of the animals per group were included in the gene assays.

7) Line 343, not clear as to why sunflower oil was used as external standard for fatty acid analysis. Sunflower contains a variety of different fatty acids, some of which are part of the de novo fatty acid synthesis pathway(s), this seems counter intuitive.

8) Line 560 to 562: " In livers of Sp supplemented groups, a significantly higher expression of GSTA4 (Fig. 5 C) and a significantly lower expression of the C-reactive protein (CRP) was observed compared to non-supplemented groups". From Figure 5F, Sp+ group seems to have higher CRP levels? Please clarify. Also, what is the impact of log2 FC scale, is this physiologically meaningful?

9) Please provide corresponding protein expression for the genes of interest in Fig 5. Gene expression alone will not be sufficient for biological interpretation.

9) In the discussion section, the main findings of the study should be introduced first.

10) There are too many details in the conclusion section. In fact, this content appears to be more appropriate to be placed into first paragraph of discussion. See comment #9

Reviewer 2 Report

The manuscript presents an interesting study. Generally, the study is well designed and described. I have only some issues:

1.            Could you put the time period of the experiment in Figure 1?

2.            Could you explain in the introduction or discussion why you expected a change in transcription data? and also in gene expression after spirulina intervention?

3.            Conclusion should be rewritten and show the most important findings.

Round 2

Reviewer 1 Report

We thank the authors for taking the time to consider comments from the evaluation. Improvements have been made, particularly with respect to the presentations of results and figures.  

Unfortunately, there are many examples of over interpretation that remain, giving the reader a view of conclusions that are not always supported by the results and dataset. The manuscript reads more like a thesis and is not concise (discussion section) raising issues that are not relevant or supported by the data and could be perceived as introducing bias. More specific examples are now provided in order to increase the clarity of these issues.
Introduction is still too long for the number of concepts needing to be raised- many sentences are repetitive, please reduce by at least 30%.  

Abstract- remove mention of non-significant data (NS) as numerically different.  

Section 3.1 results- move to the methods section as this. is part of the study design.  

Section 3.2- do not report numerically higher results if they are not statistically different (that is the whole point of statistics). Authors cite this three times in this section. Please remove.  

Results section quoting 'biological function' associated with gene expression. Please avoid interpreting the 'biological function' as the physiological modification - this is simply an association with a metabolic pathway.  Protein expression and the activity of such pathways have not actually been measured and should not be viewed as changing. Please remove.  

Section 3.5 do not report numerically higher results if they are not statistically different (that is the whole point of statistics). Please remove. Also, do not include opinion or interpretation of results in this section.  

Section 3.6 same comment here as for 3.5. The description of results are also very long winded and not needed. Reducing the commentary of 'numerically higher' data and/or trends should be removed.   

Section 3.7 No need to describe data that is non-significant (NS). Focus only on statistically significant results. Same comment for results in Fig 7.  

Section 3.8 same comment as for 3.5-3.7. Do not describe data that is NS.   Discussion. Do not discuss NS data as numerically different.   

Discussion section 4.1.1 very repetitive, reduce by 50%  

Discussion section 4.1.2 do not discuss NS data as different. Please remove. Reduce this section by at least 50%. Focus on the most important data and do not over interpret. Please avoid speculation.  

Discussion section 4.1.3 do not discuss NS data as different. Please remove.  

Discussion section 4.1.4 example whereby the section is not required due to speculation.   

Discussion section 4.2.1 This appears to be speculation and not supported by the results. please reduce to a small paragraph contains several sentences only.  

Please include a limitations section.  

Conclusion is far too long, reduce by 50% and do not be repetitive or re-state findings. 
